# Reinforcement Learning Made Affordable for Hardware Verification Engineers

**DOI:** 10.3390/mi13111887

**Published:** 2022-11-01

**Authors:** Alexandru Dinu, Petre Lucian Ogrutan

**Affiliations:** Department of Electronics and Computers, Transilvania University of Brasov, 500036 Brasov, Romania

**Keywords:** digital design, functional verification, reinforcement learning, automation, software system, stimuli generation, epsilon-greedy algorithm

## Abstract

Constrained random stimulus generation is no longer sufficient to fully simulate the functionality of a digital design. The increasing complexity of today’s hardware devices must be supported by powerful development and simulation environments, powerful computational mechanisms, and appropriate software to exploit them. Reinforcement learning, a powerful technique belonging to the field of artificial intelligence, provides the means to efficiently exploit computational resources to find even the least obvious correlations between configuration parameters, stimuli applied to digital design inputs, and their functional states. This paper, in which a novel software system is used to simplify the analysis of simulation outputs and the generation of input stimuli through reinforcement learning methods, provides important details about the setup of the proposed method to automate the verification process. By understanding how to properly configure a reinforcement algorithm to fit the specifics of a digital design, verification engineers can more quickly adopt this automated and efficient stimulus generation method (compared with classical verification) to bring the digital design to a desired functional state. The results obtained are most promising, with even 52 times fewer steps needed to reach a target state using reinforcement learning than when constrained random stimulus generation was used.

## 1. Introduction

The current complexity of integrated circuit designs makes it difficult for people who have not been deeply involved in the development process to understand the details of their functionality. However, for example, teams working in hardware design or functional verification (FV) often need to integrate multiple third-party-developed modules into their products, make them work for specific needs, and bring them into borderline situations. In these cases, reinforcement learning (RL) mechanisms (which represent a special category of artificial intelligence algorithms) can be used to find the shortest sequences of stimuli that need to be passed to the inputs of the design to make it achieve a target behavior from different initial states. 

Currently, most verification engineers are not familiar with using artificial intelligence (AI) techniques to automate their work because AI is not involved in the classical verification process. For this reason, this paper aims to facilitate the adoption of automation techniques using RL by providing both examples of agent configuration and examples of agent performance analysis. This paper can be seen as a detailed guide to approaching RL algorithms from a practical point of view, in which each configuration possibility is accompanied by examples, and the influence of configuration parameter values on the training process is evaluated. Although this paper is dedicated to verification engineers, it may also be useful to other RL practitioners, as it integrates different modern means and original methods that greatly improve the quality of the training and inference processes (e.g., integration of the epsilon-greedy concept, saving of low-performance agents). The second objective of the present work is to demonstrate that, by training agents using RL mechanisms, intelligent software reference models can be created for hardware projects. These models, represented by agents that have learned how to drive a design under test (DUT) to achieve the desired behavior, can be used to generate sequences of stimuli needed to bring a digital project into the desired state of functionality. This paper demonstrates that a training environment combined with intuitive analysis mechanisms provides results that go beyond the classical method of generating the stimulus sequences (called constrained random generation). The generated sequences are remarkable for the small number of steps they are composed of and are obtained with little effort (once the entire system is set up), both on the part of the human operator and computationally. 

The main contributions of this work are the following:Several configuration possibilities of an environment based on RL algorithms’ principles are studied in order to help RL practitioners to understand how RL works and how to adapt the RL mechanisms to their projects; for almost each configuration parameter, tables and images are employed to deeply assess the optimality of proposed configuration valuesImplementation of a software system that can control hardware simulations in order to allow RL algorithms to communicate with a DUT during training and inference processesDevelopment of a powerful mechanism based on RL that can achieve better results than the classical way of performing functional verification considering generation of stimulus sequences needed to bring a DUT into a desired state.Proposal and testing of two novel mechanisms of saving RL agents during the inference process

The rest of the paper is organized as follows. Section 2 describes the software system designed to train RL agents using either a reference model or the simulator that stimulates the DUT. Section 2 also provides background information on reinforcement learning mechanisms. Section 3 provides extensive details on the possibilities of configuring the training process and exemplifies the analysis of the results by various means. Section 4 reviews the main results, and Section 5 concludes the paper.

## 2. Related Work

The introduction of AI in hardware functional verification is not new, being an important trend adopted by several companies in the integrated circuit (IC) industry [1,2,3,4,5,6]. In this context, reinforcement learning (RL) is proving to be useful in creating reference models for digital designs, to mimic their functionality, to learn their behavior. These models are then used to generate the stimuli needed to achieve the desired functionality in an optimal time. Reinforcement learning involves an action- and reward-based training process [7]: the trained agent interacts with the digital project by providing stimuli and initiating simulations. It also reads the DUT outputs. A significant reduction in IC design time can be achieved by using artificial intelligence mechanisms in FV, as the latter consumes about 70% of the resources required in IC development [8,9]. A very common concept in the industry to migrate towards functionality-aware stimulus generation is coverage-oriented test generation, which addresses many applications of artificial intelligence algorithms in general [10,11] and reinforcement learning in particular [12].

As can be seen in the literature, there are several approaches to use reinforcement learning for automating the verification process.

One of the most well-known experiments using reinforcement learning is presented in [13]. It aims to recreate in simulation hard-to-reach scenarios in the functionality of a digital design to increase functional coverage during the verification process [14]. This target is also addressed by many research groups from all over the world [15]. The present work has the same goal and provides the possibility to train an agent to achieve any functional state of the design used as a case study (which is not necessarily a corner-case functional situation).

The work in [12] proposed an approach for the automation of the verification process that combines both RL and other machine learning (ML) algorithms. The results of this work are observed on three directions: coverage fulfillment automation, approaching very-hard-to-meet functional states using a near-miss tracking technique, and uncovering bugs by trying to reach a specific functionality of the DUT. The authors of [12] claim that, using RL, tests generated by their approach were able to achieve higher coverage values than a state-of-the-art test generator based on genetic algorithms presented in [16]. The study in [17] uses a training policy based on the *k*-nearest neighbor (K-NN) ML algorithm, approaching deep reinforcement learning (DRL) domain. The authors of [17] show how trained agents using RL mechanisms can achieve a higher coverage value compared with tests written by engineers, given operations performed on a DRAM memory. One example in [12] and the examples in [17] deal with memory-related verification. The present paper aims to enable the automation of functional verification with RL for a wide range of devices with dynamic functionality. 

The study in [18] presents a library that provides functions that can be used to automate coverage collection when working with a testbench written using cocotb [19]. The usage of a library is exemplified over two examples, where stimulus sequences are generated to automatically trigger some previously defined events (one example is related to deep learning accelerators, and the other example aims to make more efficient the verification of a network-on-chip module based on an AXI protocol, which can support a straightforward communication between modules of a system-on-chip [20]). In short, [18] aims to generate stimulus sequences to bring the DUT into a certain functional situation, not necessarily focusing on finding the shortest sequence to achieve this target. Different from [18], the approach presented here can work with ordinary testbenches written in hardware description languages (HDL).

The study in [21] presents a software system that consists of an exhaustive analysis of possible paths to a target state of an electronic circuit. These paths are further analyzed, and their size is reduced. This approach provided good results, but consumes a significant amount of resources, requiring both parallel computing capacity (for running multiple iterations of RL algorithms in a shorter time) and memory for storing the sequences of stimuli found. The currently developed approach might be easier to understand and implement, as it saves only a vector of weights (one weight for each group of states) per trained agent, and based on this vector, the shortest path to the target state is generated. The work in [22], where the RL algorithm is combined with the use of neural networks, uses an algorithm adapted after the Q-learning algorithm. In the current work, another algorithm from the same family (belonging to the temporal difference method) is used: semigradient temporal difference (SGTD).

Reinforcement learning is also used at other stages of an integrated circuit design. For example, in [23], RL is used to automatically determine the size of an electronic device based on the size of its basic components and its specifications.

Considering Table 1, several professional RL-based mechanisms are already available in the literature. However, the present work comes to fill some gaps left by the reviewed works. 

Given the reviewed works, the current paper provides the most detailed analysis of the configuration possibilities for an RL-based functional verification automation system. This aspect makes it the most suitable for engineers who need a quick transition process from classical to more automated verification. The easy adoption of the presently proposed system is also encouraged by the possibility to create a software reference model for certain functionalities of a DUT. In this way, the training process becomes considerably faster and more cost-effective compared with situations where hardware simulations are run at each RL training step. Additionally, the software system created and presented in this paper is able to work with any common register transfer logic (RTL) testbench, which means that verification engineers can integrate their legacy code into the automated environment. 

Most approaches in the literature (and in Table 1 as well) address functional verification automation focus on coverage fulfillment. The present work focuses on another aspect, generating sequences of stimuli for achieving a DUT’s target behavior, performing that in a more cost-effective way compared with the work presented in [21] (given the needed resources: in the current situation, the training was performed using only a laptop and without running several processes in parallel).

Additionally, another gap filled by the current work is related to improving the inference process (none of the reviewed works address this topic). The novelty of the current work lies not only in providing details on two useful mechanisms for using poorly trained agents during inference process, but also in implementing and testing an easy-to-adopt software system that can successfully interface the hardware simulator with the implementation of RL algorithms. A great advantage is that this system can be more easily understood and adopted by verification engineers, given their general knowledge, than many similar variants in the literature. For this reason, the use of the SGTD algorithm is a welcome feature of this paper, as it incorporates in an easy-to-understand way the most important features of an RL algorithm.

## 3. Materials and Methods

### 3.1. The Software System

To train an agent using reinforcement learning techniques, it is necessary to establish interactive communication between the simulation tool that activates and stimulates the digital module and the software environment. In the current situation, a script issues bash commands that directly call the hardware simulator. This program also creates files containing input data for the digital design, which are then made available to the simulator. After each DUT simulation completes, the same software environment reads the text files containing the DUT output and extracts the information of interest. Given Figure 1, the software code is divided into two parts: the code that contains the reinforcement learning algorithms, represented by the label *RL Agent*, and the code that creates the stimulus files to be read by the simulator, issues system commands to start the simulations, and reads the output after the simulation has stopped, represented by the label *Simulation Wrapper*.

During the training process, the code containing the reinforcement learning mechanisms provides the DUT, via the simulation wrapper, with the input stimuli required for the digital design to perform the operations. Once the stimuli are received, the simulation wrapper issues the run command to the simulator (by running a script that calls the *run.do* file shown in Figure 1, providing it with all the necessary information). In the current work, the ModelSim^®^ simulator was employed. However, the scripts can be adapted to work with any available simulator on the market, regardless of the operating system. The operation of the simulator is controlled by the simulation wrapper component. Once the simulation is complete, the wrapper reads the simulation report stored in the *transcript.txt* file and sends to the RL agent the state the DUT has reached at the end of the simulation, the calculated reward (which shows how close the DUT state is to the target state), and the information that tells whether the DUT has reached one of its final states (in some cases, it is possible for the DUT to reach more than one state, but only one of them is the desired final state). Based on the information received, the agent updates its “memory”, learning what the effect of its last action was, given the previous state. In the process of training, the agent can choose its next action either randomly or according to a predefined approach (e.g., epsilon-greedy method). The training process of an agent consists of several episodes (the iterations of the algorithm). Each training episode ends when the DUT reaches a terminal state.

In contrast to the training process, at the inference stage, the RL agent generates actions based on the experience gained in the training process, and the reward signal is ignored (Figure 2). A well-trained agent will reach the target state in a small number of steps, regardless of its initial state. A poorly trained agent will need a large number of steps to reach the target steps or even loop through nonterminal states indefinitely. This paper aims to explain how to create healthy training processes for RL agents. Furthermore, if the training process has failed to create the best-performing agents, this paper shows how these agents can still be employed. 

A DUT consisting of a basic accumulator with two operands (operand A is loaded with the value of the result signal after each performed operation) was used as the object of the present study (Figure 3). 

Its inputs and outputs are described in Table 2. This DUT has also been used to test design and verification automation mechanisms in [24,25]. The problem proposed in this paper is to make DUT reach a certain state (which is equivalent to generating a certain value on the “result” signal) by performing a minimum number of operations, whatever its initial state is. Operand B is always tied to the value “2”. For the current situation, each state of the DUT is represented by the value of the “result” signal. To achieve the proposed goal, reference models were trained using reinforcement learning mechanisms. These models, by choosing a sequence of appropriate values for operand B and the operation code, were able to make the DUT obtain the desired output.

In case of other DUTs, engineers must assign each DUT state a set of output signals or another indicator that can be read after a simulation has been run. In this case, it is possible to feed the RL algorithms the information needed to correctly construct the correlations between the input stimuli and the sequence of DUT states.

### 3.2. Theoretical Background

Reinforcement learning is one of the most important machine learning algorithms, along with supervised learning, unsupervised learning, transduction, and so on [26]. This technique aims to train a software model (called “agent”) to give it a certain functionality. This functionality lies in generating desired logical outputs based on either its state or its state and its inputs (it can operate either as a Moore machine or as a Mealy machine [27]). The training process consists of supplying the initial model (which is described by an array of values that are called “weights”) with random stimuli over several iterations and rewarding the agent based on the “actions” it performs (which, in hardware, are thought of as generating values for certain signals). The reward function is constructed to cause the agent to reach a certain desired functional state, which could be characterized by a combination of its output signals. At each training stage, after having put certain values on its outputs and having received its reward, the agent “understands” how appropriate these values were and updates its “memory” (the values in its weight array) accordingly. For a better understanding, the training process is similar to the process of learning to walk: a child’s brain learns how to send signals to the leg muscles to maintain balance. Perfect balance comes after many attempts and after the child falls or loses balance several times while trying to walk. Returning to the scientific explanation, the agent must maximize its long-term reward by achieving a sequence of consecutive generations that will bring it to its target. After several iterations of training, from which it understands the effects of its actions according to its current state, the agent becomes able, regardless of the state in which it is initialized, to choose the stimuli necessary to get as close as possible to the target state. Transposing this to hardware functional verification, an agent may become able to make the DUT reach a particular functional state by successively supplying it with stimuli. The quality of the learning process is highly dependent on reward functions, state grouping, and other configuration elements whose explanation is the main focus of this paper. 

There are several ways to apply reinforcement learning to achieve a goal. Depending on the actions the agent can take, the number of states, and other relevant parameters, an appropriate algorithm can be chosen. In the current work, the semigradient time-difference(0) (SGTD(0)) algorithm was used, which belongs to the category of approximation solution methods, as defined in [28]. During the training of an agent using this algorithm, the state-value function $v_{\pi }$ has to be approximated. This function will guide the agent during the inference process to choose the appropriate action given its current state. The goal of training a model is to make each state value move towards the target value. In most cases, the target value is unknown, and its value is the target of the training process. By implementing a reward system and letting the agent move from one state to another, the model learns what is the desired behavior of the function to be approximated and updates its weights accordingly.

The algorithm used in this paper for updating the weights used to approximate the state-value function $v_{\pi }$ belongs to the SGD (stochastic gradient descent) family of algorithms. As mentioned in [28], “gradient descent” methods take their name from the equation used for updating the vector of weights w from one iteration to the next, which contains the gradient (the column vector of the partial derivatives of $\hat{v}\left(S_{t},w_{t}\right)$ differentiated by each weight element at a given time). The term “stochastic” indicates that the example selected for updating the weights vector w could be chosen randomly. 

When using the linear function approximation in SGD, the characteristic vector becomes the “gradient of the approximate value function with respect to w” [28], which makes the mathematical calculation of the weighting vector update very simple, as can be seen in Equation (1), where all the necessary information related to the weighting vector calculation is given:(1)$$w_{t+1}≐\;w_{t}+\left[\begin{array}{c} \begin{array}{c} 0\\ \ldots \\ 0 \end{array}\\ \alpha \left[R_{t+1}+\gamma \hat{v}\left(S_{t+1},w_{t+1}\right)-\hat{v}\left(S_{t},w_{t}\right)\right]x_{i}\left(S_{t}\right)\\ \begin{array}{c} 0\\ \ldots \\ 0 \end{array} \end{array}\right],$$
where $w_{t+1},\;w_{t}$ represents the column vector of weights (“memory” or “experience” of the agent) at two consecutive training steps; γ is called the discount factor: this parameter makes the agent take into account not only the reward it takes by going to the next state, but also how valuable the next state is; and *α* represents the step-size parameter that specifies how much the model is influenced by the last accumulated experience. If *α* has a value closer to 1, the agent reaches the better behavior faster, but it cannot settle to the most desirable behavior even if it is trained for a long time, because it continuously updates its weights and thus oscillates around the best behavior. To solve this problem, the task of the programmer is to sample the vector of weights several times during training and keep for inference the best-performing model (each model being characterized by the matrix of weights).

## 4. Results

In order to practically evaluate the influence of different coefficients used in reinforcement learning and to find the best training configuration, the values of the training and inference parameters were changed several times during this study. The experiments started with easier targets and, as the training process improved, the severity of the achievement conditions was gradually increased. In order to be able to compare the trained agents, a series of starting states was defined, as seen in Equation (2). At inference, agents were initialized in each of the starting states, and agent performance was considered inversely proportional to the number of steps required to reach the desired end state from each of the given starting states.
(2)start state ∈{248, 34, 144, 50, 0, 90, 45, 255, 15, 69,199}

### 4.1. Influence of the Randomness on Agent Training Processes

During performed experiments, the values of operand A and the result are 8 bits wide; therefore, there are 256 different states that the agent can reach. Fifty groups of states have been created, each group containing 5 states. Therefore, the values for states 250–255 (the states that do not belong to any group) will remain 0 and will not be reached during the training process. Although it seems inefficient to omit some states from being associated with a weight, it can be seen from Table 3 that the inference process can still give good results compared with using random actions (random actions meaning random choice of states until the agent reaches the desired terminal state). However, from Section 4.4 onwards, all states will be contained by part of a group (e.g., for 256 states, where each group has 5 elements, there will be 52 groups).

All agents have been trained, setting the desired terminal states to “61”, “62”, or “63”. Dangerous terminal states were set to “1”, “2”, and “3”. During the training process, when the agent reaches a desired terminal state, it receives a positive reward (i.e., +1). When it reaches a dangerous state, it receives a negative reward (i.e., −1). Finally, when it reaches a nonterminal state, it receives a neutral reward (i.e., 0). This reward scheme is shown in Figure 4.

Three agents were trained using the described configuration over 1000 iterations. In each iteration, the agents started from a given state and, by issuing random actions, moved through state space until it reached one of the set terminal states. Agents behave differently because the seed that influences the pseudo-random generation of actions is unique to each of them. To speed up the training process, a Python program that implemented the DUT functionality was created and used. In this way, the training time for each agent took about 1.8 s. After the training processes, the agents were tested by being put in each of the initial states in Table 3 and were allowed to find the best path to the desired final state. In this way, the performances noted in Table 3 were obtained. To highlight the performance of the trained agents, it was tested in how long an untrained agent, choosing with equal probability an action independent of its current state, reaches the desired final state. 

One of the conclusions drawn from this section is that different agents that are trained using the same configuration may perform differently if the value of the “seed” parameter is changed. Therefore, after choosing a training configuration, it is worthwhile to train several agents and choose for inference the best-performing one according to the evaluation criteria. 

Another conclusion concerns the large difference between trained and untrained agents. As can be seen from Figure 5, all trained agents gave a high value to states close to the target state, states close to twice the target state, and so on. Similarly, they gave a low value to the states whose number was closer to the undesired terminal state. These lower-numbered states also consumed a lot of training steps, because it is difficult to go from small numbers to larger ones using operations where the second operand was always “2”. In contrast, untrained agents wasted a lot of time jumping from one small number to another, and this often took them away from the desired end state.

### 4.2. Choosing the Best Reward Scheme

Semigradient TD(0) is the algorithm used to update the weights. Each weight of a group is updated each time the agent reaches a state within that group. During the current training process, the agent always chooses random actions and estimates the value of the reached states based on the reward received and the value of the next state that could be reached. However, since the next state is chosen randomly, the agent can hardly estimate the correct value of the reached state. As a side note, the notions of agent and run will overlap throughout this paper, since an agent is trained by issuing a run of the reinforcement algorithm.

In the functional specification of an IC, information about the situations in which the DUT is expected to operate is found. Additionally, scenarios of limit situations are specified, which must be recreated in the simulation to prove that the DUT continues to perform well in these situations. Moreover, the DUT should not be able to misbehave whatever stimuli it receives. Therefore, in many real cases, there are certain goals that the DUT must achieve, but it is not expected that the DUT will end up in dangerous situations during operation. If necessary, however, even from the phase when the DUT receives stimuli, illegal or dangerous combinations of stimuli are filtered out in the simulation and inactivated during the operation of the real project. Therefore, in this project with the focus on creating a reference model of a DUT, dangerous terminal states have been removed (Figure 6 shows the new concept of the training environment), and new agents have been trained again using the same desired terminal states: 61, 62, 63. The purpose of this analysis was to see whether agents trained without dangerous states performed better than agents that encountered dangerous states (in which the agent received less reward than in the usual states) during the training process.

As seen above, several reward schemes were used in the current project to find out which configuration gives the best results: which type of reward leads to the agents with the highest accuracy.

The following reward schemes are compared below:Both the dangerous terminal state and the desired terminal state consist of three consecutive states. Dangerous terminal states are “1”, “2”, “3”, and desired terminal states are “61”, “62”, “63”. When the agent reaches dangerous terminal states, it is rewarded with the value “−1”. When the agent reaches the desired terminal states, it is rewarded with the value “+1”. When entering all other states, the agent is rewarded with the value “0”.The desired terminal states are “61”, “62”, “63” and there are no dangerous terminal states. When the agent reaches the desired terminal states, it is rewarded with the value “0”. When entering all other states, the agent is rewarded with the value “−1”.The desired terminal states are “61”, “62′, “63”, and there are no dangerous terminal states. When the agent reaches the desired terminal states, it is rewarded with the value “+1”. When entering all other states, the agent is rewarded with the value “0”.

In the current work, the third reward scheme gave the best results, creating agent models that reached the desired terminal state faster in comparison with the agents trained using the other two reward schemes, as can be seen in Table 4. This is the reason why the third reward scheme was selected to be further used during the current work. In the current work, the third reward scheme gave the best results, creating models that reached the desired terminal state faster compared with agents trained using the other two reward schemes, as can be seen in Table 4. This is why the third reward scheme was selected for further use in the current work.

When the agent failed (during the inference process) to reach the desired final state in fewer steps than the number of possible states, it means that the agent started visiting the same state more than once. This behavior proves that the agent was poorly trained. In order to save even poorly trained agents and to use them during the inference process, the marking visited state (MVS) mechanism has been introduced. A similar approach has been used in implementations of different algorithms (e.g., [29,30,31]). It consists of giving the value “−100” to each state visited during the inference process immediately after visiting it. Each time a state is visited, its value is updated to “−100”, meaning that the agent must never return to it. As an important note, agent weights saved after training are not changed, but only the state value for the current inference process. When another inference process starts, the agent will have again the value of the weights that were updated during training. Therefore, given that agent weights have values between “−1” and “+1” (see Figure 5), or even between “0” and “1” if negative rewards are avoided, the agent will prefer to reach only unvisited states on its way to the desired terminal state, and loops created in cases where the agent has returned to a state already visited are interrupted. A poorly performing agent can be easily detected if the number of training steps exceeds the number of available states. These situations are marked in Table 4 with “>256”. To better assess the quality of trained agents, a fourth column was added to the table, in which a completely untrained agent issued random actions until it reached the desired end state. The last column in Table 4 was created using Equation (3), following the intuition that: (1) if the initial state is greater than the desired terminal states, by consecutively subtracting 2 from the current accumulator value, it is impossible not to reach the desired terminal state; and (2) if the initial state is less than the desired terminal states, by consecutively adding 2 to the current accumulator value, it is impossible not to reach the desired terminal state.
(3)$$steps\;number=INT\left(\frac{ABS\left(start\;state-desired\;terminal\;state+1\right)}{operand\_B\_value}\right),$$

The agents whose performance can be seen in Table 4 were configured using the parameter values found in Table 5. The time required to train each agent depended on the reward scheme. The average training time for agents created after the first reward scheme was 1.8 s; for agents created after the second reward scheme, it was 7.2 s; and for agents created after the third reward scheme, it was 6.5 s. 

### 4.3. Mechanisms for an Efficient Inference Process

This marking-visited-states mechanism is only required for poor-quality trained models. This can be seen in Figure 7, where the application of the MVS concept definitely helped to reach the desired terminal states (number of steps = “−1” represents that the agent could not reach the desired terminal state in a reasonable time; therefore, it was interrupted after executing a number of steps equal to the number of states in the environment).

As probably already noticed, at inference, only the value of the states visited at each of the agent’s steps was changed to “−100”. However, in the training process, the states in a group had the same weight, so they had a certain value. The idea that consecutive states have the same value can also be used at inference. Therefore, if a state has already been visited and was not the final state, it can be assumed that the final state is not yet close, and the close neighbors of the visited state (adding and subtracting “−1” from the value of the visited state) can also be given the value “−100” to avoid visiting them in the future in the current inference process. This approach is somewhat similar to the functionality of Dijkstra’s algorithm [32]. We call this approach the extended-marking-visited-states (EMVS) mechanism and apply it by ensuring that neighboring states that will also be marked as visited are not the desired terminal states. The EMVS mechanism is mathematically represented by the steps in Equation (4).
(4)$$for\;each\;visited\;state:\;\;i=index_{visited\;state}\;state_{i}=-100,\;state_{i+1}=-100\;state_{i-1}=-100,\;\forall \;state_{i}\;\in \;S,\;state_{i}\;\notin \left(\mathrm{desired}\;\mathrm{terminal}\;\mathrm{states}\right)$$

The results of applying the MVS method, the EMVS method, and the random method for action selection in the inference process are shown in Figure 8. It should be noted that when the initial state was equal to 0 and 69, EMVS outperformed the MVS mechanism.

### 4.4. Analysis of the Influence of the Discount Factor on the Training Process

In the inference process, even for the third reward scheme, it was observed that low-quality trained agents were obtained at the end of the training process. Consequently, infinite loops appeared when the agent tried to reach the desired terminal state. For example, looking at the arrows in Figure 9, where the initial state of a training iteration was 248, the agent reached state 124 by division by 2 and then continued to alternate between states 124 and 122 because the values of the desired terminal states (61, 62, or 63) that could be reached by dividing 122 or 124 by 2 were poorly estimated. One of the possible reasons why the value for state 124 and neighboring states was estimated to be higher than the true desired terminal state value is that in the training process, states 122 and 124 were reached often, and their value was increased several times. In the semigradient algorithm TD(0), the value of a state is updated based on the value of the next estimated states. Therefore, the values of state 124 and neighboring states were also increased a lot on behalf of the desired terminal state.

The training parameter that can be used to solve the problem that the estimated value of the desired terminal state is not the largest is the discount factor (γ). If its value is reduced, the value of a certain state will rely more on the reward received for advancing to another state than on the estimated value of the next state (bootstrapping influence is reduced). To check the impact of the discount factor on agent accuracy, several agents are trained using the discount factor values from Equation (5), meaning that γ takes all values from 0.3 to 1.0, with a step of 0.1.
(5)γ ∈{0.3:1.0:0.1}

To evaluate the impact of the discount factor on the quality of the training process, the general training setup in Table 6 was used for the learning processes of 10 agents trained using the same configuration, but with a different seed. Training one agent (over 3000 episodes) took approximately 8.5 s, using the Intel^®^ Core™ i7-4810MQ CPU configuration @ 2.80 GHz, aided by 16 GB DDR3 RAM. Additionally, running RL training processes on the CUDA GPU was attempted, but slower performance was achieved. One explanation could be that, unlike supervised learning, where a huge amount of data has to be correlated with its labels and this process is run over thousands of parallel processes, in reinforcement learning, each action of the agent is processed individually, and the agent can choose the next action only after obtaining a result from the previous one. Thus, the additional cost of transferring data to the GPU [33] several times during the training process takes its toll by introducing higher latency compared with transferring data from memory to the CPU.

For each discount factor, 10 agents were trained. The seed used in the training process for each agent is equal to the agent index: for the 1st agent, seed “1” was used; for the 2nd agent, seed “2” was used, …, for the 10th agent, seed “10” was used. The seeds contribute to the random generation of agent actions during the training process and to the choice of the agent’s initial state. Given the results in Table 7, the conclusion in the section of this paper “Section 4.1” is again proven: the quality of the agent training process can be influenced by the seed. Therefore, it is necessary to train several agents, using for each of them a different seed, and to choose the best-performing agent to be used in the target application (e.g., as a reference model).

Considering the number of steps required for each agent (the label of each agent is found on the run number axis) to reach the desired terminal state starting from each available state in the environment, the quality of the fourth trained agent (named “run_4”) is easy to observe, since a gap is present at each image where the execution number equals 4.

For a better understanding of the results in Table 7, Figure 10 shows the dependence. This means that agent 4 took fewer steps to reach the desired terminal state, whatever its initial state was. In conclusion, if one would like to use a trained agent to create a machine learning application, agent 4 (seed must be 4) is one of the appropriate choices.

From the results in Table 7, it can be concluded that, in the current situation, it is more lucrative to use smaller discount factors for updating weights using the semigradient TD(0) algorithm. Additionally, as can be seen in Figure 10, when agents were trained with a discount factor having the values 0.9 or 1.0, the overall performance of the agent decreased (the average number of steps required increased, as can also be seen at the upper limit of the steps’ number axis). A completely unconstrained episodic task reduces too much the importance of the reward received by the agent after issuing an action, leading to problems similar to those captured in Figure 9: the value of intermediate states becomes larger than the value of desired terminal states.

### 4.5. Optimality of State Grouping

The semigradient TD(0) algorithm can handle environments containing hundreds of states. In order to speed up the training process, the states are grouped, each group having the same weight. A simple but efficient approach is to form equal groups of neighboring states [34].

Given the current work, if operand A is 8 bits wide, the agent can reach 256 states. To see what the advantage of grouping states is, the states are grouped into:50 groups, each state group containing 5 states;250 groups, each group containing 1 state;5 groups, each group containing 50 states.

Each state grouping focuses only on the first 250 states out of a total of 256 states. Therefore, the weights for states 250–255 will always be 0, not reached during the training process.

The performance of the agents in terms of the clustering approach, for several values of the step-size parameter, can be seen in Figure 11. In the present situation, the training process has been performed under the condition that, at each episode, the agent starts from the same initial state.

In the first three cases analyzed below, the value of the step size was kept constant, which is not recommended because it cannot be ensured that the weight vector will converge to the true state values: Note that both convergence conditions are satisfied for the sample-average case, $\alpha_{n}\left(a\right)=\frac{1}{n}$, but not for the case of constant step-size parameter, $\alpha_{n}\left(a\right)=\alpha$” [28]. Only in the fourth case, the step-size parameter is determined according to Equation (6).
(6)$$\alpha_{n}\left(a\right)=\frac{1}{n},$$
where *n* represents the “*n*th selection of action a” during the training process.

Although there are the issues mentioned above that might affect the training process, it can be seen in Figure 11 that agents manage to reach the desired terminal state in each episode in a reasonable number of steps.

In Figure 11, the axes labeled “Number of groups of states” and “Steps number” are numbered logarithmically to better understand the difference between the trained models. Therefore, there is a small but significant difference between the cases where the number of groups was equal to 50 and 250, showing that grouping of states can bring real benefits in many cases when applied during the training process of RL agents. Additionally, adding more than one state in the same cluster proved to be very inefficient: the behavior of states that are not close neighbors differs greatly between them, and the update of the cluster weight when the agent arrives in one state in the cluster can be cancelled out by an opposite update of the same weight when the agent enters another state in the same group. In conclusion, it is not beneficial, during the training process, to group states with opposite effects. To limit this possibility, groups should contain an appropriately limited number of states. In the current situation, placing five states in the same group has proven to be an optimal decision.

Additionally, in Figure 11, a white line can be clearly seen near the desired terminal states. The reason is that, in order to create a more accurate training process, when the agent starts from the desired terminal state, the number of steps required to reach the desired terminal state is set to 0. The results in Figure 11 were obtained by configuring the training process with the parameter values in Table 8.

In conclusion, unless the step size is equal to 0.01, creating groups of 5 states improved agent performance. Therefore, it is worth looking for possibilities to group states even in environments with a relatively small number of states (such as this accumulator in the present paper, which, for the 8-bit wide operand A, can reach 256 states). In Table 9, configured according to data in Table 8, the last two situations described visually in Figure 11 are summarized numerically.

It can be observed that the concept of state grouping can streamline the training process. However, depending on the number of states in each group, the model may improve its learning or often overwrite its past memory with a new unexpected experience. The latter situation occurs when states with opposite values (effects) are put together in the same group. Therefore, the value of the group will fluctuate strongly when these states are accessed. To find the best number of states that should be part of a group, several trials should be performed, as exemplified in this section.

### 4.6. Choosing the Best Value for the Step-Size Parameter

The step-size parameter shows how much an agent learns at each episode step. From a mathematical point of view, given Equation (1), the step size α limits the weight update with the information learned in each training iteration. If the step size has a larger value (close to “1”), agents learn faster, but after they gain experience, they do not rely on the acquired experience, but always update their weights according to the last things learned. If the step size has a smaller value, agents slowly update their experience (which is mathematically represented by the weights matrix) with the newly learned information. According to [28], a recommended method for updating the step size value is to use Equation (6), whose benefits can be seen in Figure 11.

### 4.7. Integration of the Epsilon-Greedy Concept

Up to this point in the training process, agents have randomly selected actions to move from one state to another, trying to reach the desired terminal state. However, a more efficient method (a more convenient action selection policy) should be to allow the agent to make use of the experience gained right from the training phase. After a few stages of training, an agent may have the opportunity to follow the most convenient path he has learned (visiting high value states) or to explore new states, hoping to find better solutions to reach the target than the paths he has already found. The epsilon (ϵ)-greedy approach always offers two options:Exploiting already-acquired knowledge, to maximize the short-term reward;Exploring of new potential possibilities to reach the target, to maximize the long-term reward.

It is essential that an agent learns a lot at the beginning of the training process and uses the experience gained in the second part of the training process. Thus, in the first part of the learning phase, the agent explores many states in the environment and figures out which ones are most valuable. Further, in the second part of the learning phase, the agent exploits the most profitable paths, and their value is increased and well separated from unprofitable states (by changing the corresponding weights). To implement this mechanism, the value ϵ is initialized with “1” at the beginning of the learning process and is decreased as the agent’s experience increases. Additionally, at each training step, a value between 0 and 1 is randomly generated: if this value is greater than the value ϵ, the agent will choose to enter the state with the best value it can reach, selecting its next action accordingly (this is the exploitation approach); if the random number is less than ϵ, the agent will choose a random action that will bring it to a state independent of the known value of that state (this is the exploration approach). Consequently, the agent will mainly choose the exploration branch while building its experience and will mainly choose the exploitation branch when it is nearing the end of the training process. This decision-making process is also depicted in Figure 12.

In the current implementation, ϵ is initialized with “1” at the beginning of the training process. During the training process, ϵ is decremented (1) either at the end of each training episode or (2) each time the agent enters the exploration branch.

In order to have a good understanding of how the ϵ-greedy method can be conveniently applied, several comparisons have been performed, as shown in the following sections. The lack of ground truth, which could show which model behaves best with respect to the reference, is compensated by training a model in different ways and selecting the best-performing instance. The number of steps required for a model to reach the desired terminal state, after being initialized in one of the environment states, is the metric used to compare the performance of multiple agents.

### 4.8. Steps for Obtaining Performance Using RL in the Automation of Functional Verification

In order to obtain high-performance RL-based agents, a number of aspects must be taken into account, which will be summarized in the current chapter.

#### 4.8.1. Defining Values for Configuration Parameters

Table 10 lists the main parameters that can be used to configure the training process of an RL agent.

The parameter ϵ initial value is the value used to initialize ϵ at the beginning of the training process of each agent. The ϵ decrement represents the value that is subtracted from ϵ each time the the decrement_epsilon() function is called in the training process. In the training process characterized by Table 10, this function is called on the exploration branch.

Currently, in each episode, agents are initialized to the same initial state. Two different agents start their training iterations from 2 different starting states. Additionally, parameter ϵ is decremented each time the algorithm reaches the exploration branch. In addition, when comparing agent performance, inference with the MVS mechanism is used.

#### 4.8.2. Training Multiple Agents Using the Same Configuration

To use the performance evaluation methods, 10 agents are trained using the configuration in Table 10. 

The training times of the agents and the epsilon value at the end of the training process of each agent are shown in Table 11. The epsilon value differs from agent to agent because each agent has its own experience (one agent learns faster, and one agent learns slower, given its random actions and the states it has reached), and the duration of the episodes does not have a fixed number of steps: the episodes end when the desired end state is reached. During the episodes, each agent accesses the exploration branch a different number of times, and thus the, parameter ϵ is decremented by a variable number of times. 

#### 4.8.3. Analysis of the Performance of the Generated Agents

A mechanism for comparing the performance of the generated agents must be implemented, and the results can be verified both numerically and visually. At the inference stage, agents are initialized into all relevant states of the environment (in this case, all states can be approached, as there are only 256 states), and the steps required until the agent reaches the desired terminal state are counted. For visualization purposes, if the agent fails to reach the desired terminal state after executing a number of steps equal to the number of states, the agent is noted on the graph to have reached the terminal state in 400 steps. This shortens the lengthy inference processes that occur when a poorly trained agent is in focus. In Figure 13, it can be seen that agents with indexes 2, 3, and 4 performed better than the other agents. The agent with index 8 performed the worst, because it often failed to reach one of the desired terminal states when starting from the other states (most of its inference processes are marked with 400 steps).

Since it is not visually possible to understand exactly which agent performed best, the average number of steps for each agent was also calculated and introduced in the last column of Table 11. Initial states for the 20 inference episodes are uniformly distributed across the range of states. The best model turned out to be agent 4, as can be seen in Table 11.

The values for each state that were learned by agent 4 can be seen in Figure 14, represented by the orange color. It should be noted that although the graphs of agents 3 and 4 differ considerably, their performance is similar. One possible reason is that the value of very important states (62, 124, 248, 32, 159) is high in both graphs.

#### 4.8.4. Changing the Training Configuration

In order to have chances to obtain a better-performing agent, the configuration of the training process has to be changed several times, and the training and inference steps have to be reassumed.

First, in contrast to the configuration shown in Table 11, the value of the “ϵ decrement” was doubled. This resulted in a very long training process: the ϵ value reached 0.1 (its minimum value) too quickly, and the still-untrained agent tried to reach the desired terminal state in the subsequent episodes, mainly taking into account its experience. However, since it was not able to accumulate enough experience until ϵ reached the value of 0.1, it reached the desired terminal state after completing many more steps than if epsilon was 0.00001. The second attempt to change the parameter aimed at reducing the value of the “ϵ decrement” to half of its previous value (from 0.00001, it was changed to 0.000005). The performance of the trained agent can be evaluated numerically in Table 12 and visually in Figure 15.

Agents 5 and 8 performed best in the current training configuration and also outperformed agents trained using the previous configuration.

Plots of the state values of agents 5 and 8 are shown in Figure 16. It should be noted that the desired terminal states (61, 62, 63) have the highest value, followed by states 124 and 248 (multiples of two and four of the desired terminal state). Agent 8 also detected states 186 and close neighbors as valuable states, since these states are multiples of three of the desired terminal states.

To better understand the influence of training parameters, other training configurations were tested. The results of these can be seen in Table 13. The ϵ decrement was increasingly reduced until, in configuration c, the epsilon-greedy mechanism was completely eliminated. Given these results, it can be concluded that the epsilon-greedy mechanism did not significantly influence the training process in the current situation, when the environment contained a small number of states. For a better overview of the optimality of different configurations of the training process, the number of states in the environment needs to be increased. This can be easily achieved given the studied accumulator by increasing the width of operand A.

By presenting some of the possibilities of using RL agents, verification engineers can have an idea of the RL potential of automation. The approach presented in this paper can also be adapted for a wide range of projects in order to create reference modes capable of generating meaningful sequences of input stimuli. 

Finally, by continuously improving the training configurations and making effective use of mechanisms to streamline the inference process (such as EMVS), agents with high performance in reaching the target state were obtained. Therefore, the agent trained according to configuration b in Table 13 achieved more than 52 times higher performance compared with the random stimulus generation (the exponent of the classical way of performing functional verification) seen in Table 4 and more than 16 times higher performance compared with the data present in the last column of Table 4 (which represents a fictitious way of bringing the agent to the target states; certainly, in complex projects, it is very difficult or even impossible to create such rules that can bring the reference model to the target states). 

## 5. Discussion

Reinforcement learning is a suitable companion in tasks aimed at accessing digital design behavior in a minimum number of steps. By using reinforcement learning, software reference models can be created for a wide variety of modules and hardware components. The quality of these models depends largely on the quality of the agent training process.

In the current study, a novel software system was designed to allow information exchange between RL agents and the simulator. For this reason, a simulation wrapper was created that was able to create input stimulus files for the DUT, issue commands to the hardware simulator, and extract simulation outputs.

To achieve the proposed goals, several RL agents were trained to make them learn the appropriate inputs that the DUT must receive to reach the target state. According to the results obtained, using RL delivers the desired stimulus sequences faster than using constrained random stimulus generation. The random generation of data achieved an average of 109.55 steps (Table 3), and the best-trained agent achieved an average of 8.36 steps. Therefore, in contrast to the random generation, constructing stimulus sequences using a well-trained agent decreased the average number of steps required to reach the target state of DUT functionality by more than 16 times. Considering Table 13, where an advanced configuration scheme was applied, one of the RL agents became 52 times more performant than its counterpart which used random stimuli.

The performance of agent training processes is influenced by the values given to the configuration parameters. According to the analysis performed, if several agents are trained with the same training process configuration, but a different “see” is used to generate the random values that were required during training for each agent (e.g., the initial state of the agents at each training episode, the actions selected during training, the value that is compared to ϵ at each training step when using the epsilon-greedy approach), the performance of the agents differs (Table 3). Therefore, after training several agents using the same training configuration, it is advisable to compare the performance of the agents and choose the best one to further use as a reference or predictor.

It has also been observed that the use of the mechanism of marking visited states during the inference process can lead to the rescue of poorly trained agents, providing the possibility to use them to reach the target state in a reasonable number of steps (Figure 7). Additionally, marking neighbors of visited states as visited can further help the agent to reach the target faster (Figure 8). Therefore, EMVS represents a novel way in reinforcement learning of rescuing poorly trained agents, introduced by this paper.

In the current work, the most effective method was to give a positive reward (“+1”) to the agent when it entered the desired terminal state and a neutral reward (“0”) when it entered the other states. It should be noted that the reward scheme has a strong impact on the training process, and it is necessary to try several reward schemes at the beginning of building a reinforcement learning application.

As well, a completely undiscounted approach (γ = 1.0) was not successful. When the value of operand B was tied to “2”, the best-performing agent configuration had the discount factor (γ) equal to 0.4. Therefore, giving a reward more than twice the estimated value of the next state, and thus reducing bootstrapping influence while the agent learns, led to a more accurate training process.

Additionally, as seen in the “Section 4.5”, state aggregation can be beneficial for the training process even when the number of states is not very large. An important point is that groups should be sized so that the risk of grouping states with opposite effects is minimized as much as possible.

It has also been observed that integrating the epsilon-greedy policy into the agent training process does not influence the training process when the environment contains a small number of states. However, according to the literature (e.g., [35,36]), this mechanism brings important advantages when the agent’s environment contains a large number of states. A study to be addressed in the future is to determine use cases where epsilon-greedy can improve the training process to obtain better-performing stimulus sequences. 

In addition, in this paper, only one example of a simple circuit whose states are to be achieved by emitting a sequence of stimuli is presented. Therefore, a direction for future research is to address designs of different levels of complexity and establish a guiding correspondence between the complexity of a DUT and the effort and resources required to train agents using RL mechanisms. In this way, verification engineers will be able to better estimate at the beginning of the work when it is worth training agents to achieve the desired stimulus sequences and when it is more convenient to use only constrained random generation, which is the exponent of the classical way of performing functional verification.

## 6. Conclusions

Reinforcement learning proves to be a welcome automation mechanism for functional verification. This method can make judicious use of computational resources, obtaining the desired stimulus sequences in a shorter time compared with the classical way of performing verification.

This paper has provided examples and hints related to the configuration of an RL agent and has exemplified some significant ways of evaluating the performance of the created agents, considering the intended goal. 

Additionally, this paper has demonstrated that agents trained using reinforcement learning tools can successfully learn behaviors of digital models by interacting with them. This advantage can be exploited in many ways (e.g., for decreasing the time needed for fulfilling the functional coverage collection during the verification process). Moreover, these agents can generate the stimuli needed to bring the digital design to a particular state much faster than by using pseudo-random stimulus generation, the latter being a representative feature of the classical way of performing verification.

Last but not least, the approaches of marking the visited states (MVS, EMVS) proposed in the current paper, in an original implementation, for saving poorly RL-trained agents allowed us to reduce the time spent during the training process. Therefore, these mechanisms support the generation of short sequences of stimuli needed for a DUT to reach the target state in a minimum number of steps. 

In a future work, a comparison between the use of the epsilon-greedy concept and the use of DNN for faster acquisition of the desired stimulus sequences should be performed. Additionally, the SGTD algorithm should be replaced by the Q-learning algorithm in the same context, and the performance obtained should be compared.

## Figures and Tables

**Figure 1 micromachines-13-01887-f001:**
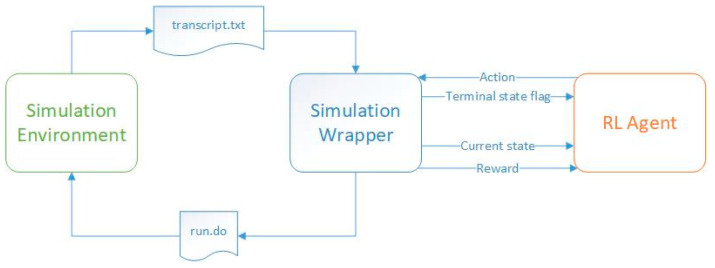
Information transferred between the code representing the reinforcement learning agent and the simulation environment containing the DUT during the training process.

**Figure 2 micromachines-13-01887-f002:**
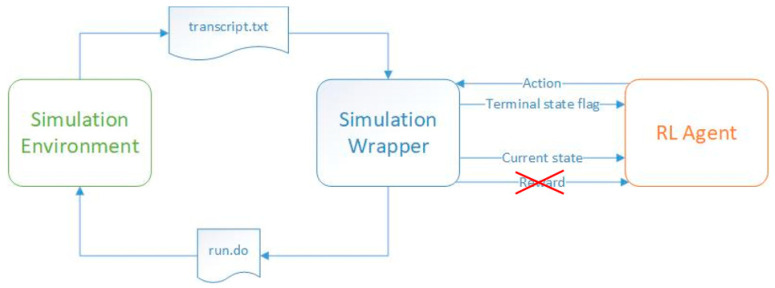
Information transferred between the code representing the reinforcement learning agent and the simulation environment containing the DUT during the inference process.

**Figure 3 micromachines-13-01887-f003:**
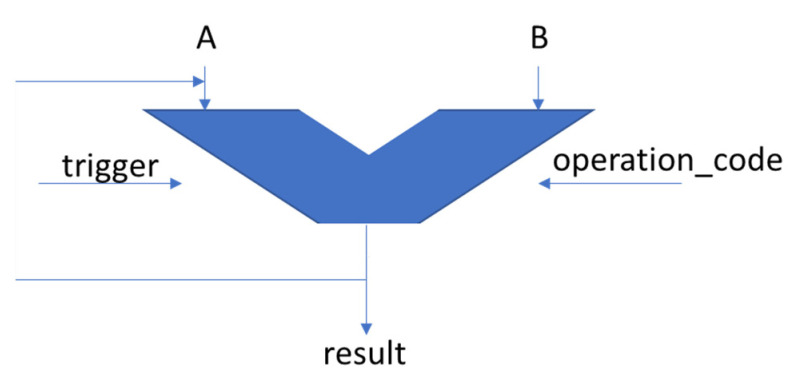
The DUT used as an example in the current paper.

**Figure 4 micromachines-13-01887-f004:**
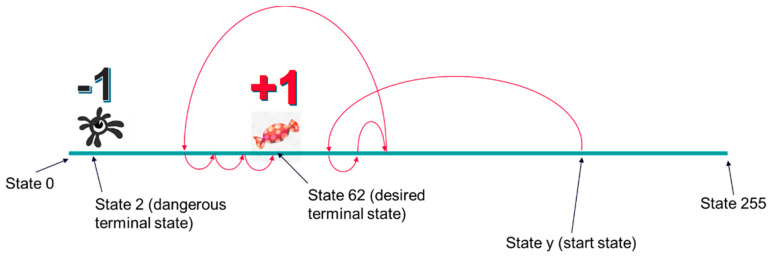
Reward scheme used in the first training setup. Only a desired terminal state and a dangerous terminal state are represented in the picture.

**Figure 5 micromachines-13-01887-f005:**
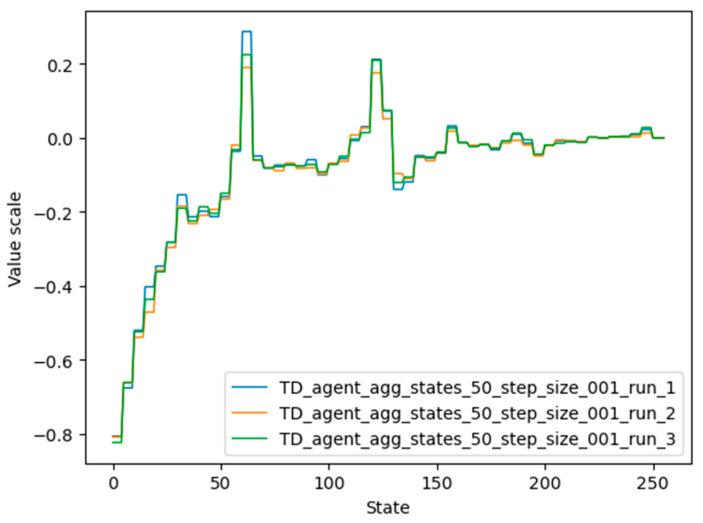
Trained agents’ “memory” used to assess the value of each group of visited states. Each group contains 5 states; therefore, each of the 5 consecutive states has the same value.

**Figure 6 micromachines-13-01887-f006:**
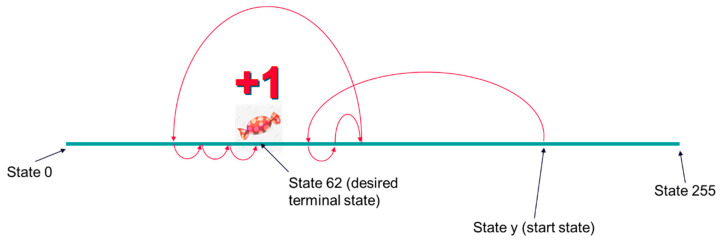
Environment created when operand A is 8 bits wide, the desired terminal state is 62, and there is no dangerous terminal state.

**Figure 7 micromachines-13-01887-f007:**
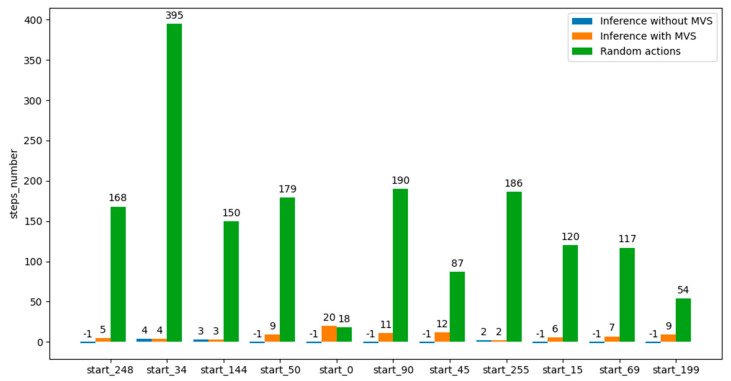
Performance graph of whether or not the MVS mechanism was applied at the inference stage for an agent trained using the third reward scheme.

**Figure 8 micromachines-13-01887-f008:**
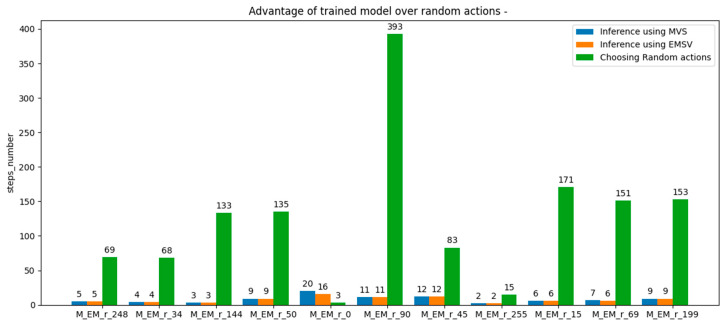
Inference process results on a trained agent model with the third reward scheme, where the MVS and EMVS mechanisms were applied one by one, compared with randomly choosing actions until the desired terminal state was reached. The last number in the description of each group of bars represents the initial state for the respective inference process.

**Figure 9 micromachines-13-01887-f009:**
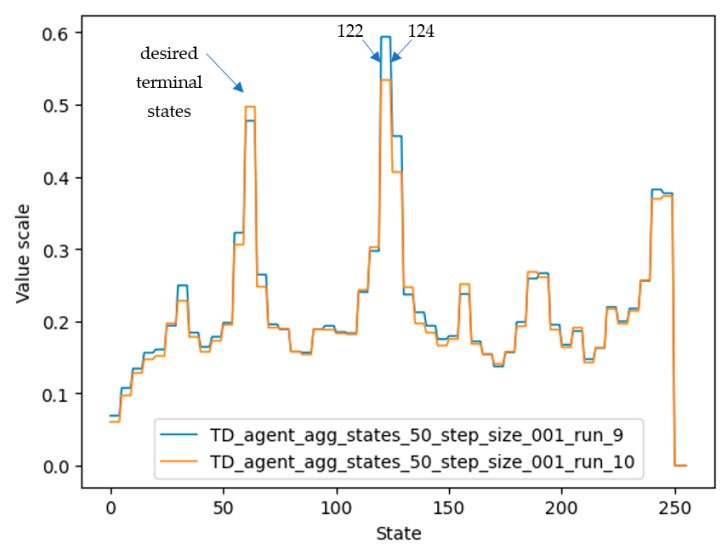
Vector of state values after training two agents using the third reward scheme. Two agents (represented by run_9 and run_10) are shown in the figure. It can be seen again that although the same training setup was used, they learned differently because they randomly selected different actions during the training process.

**Figure 10 micromachines-13-01887-f010:**
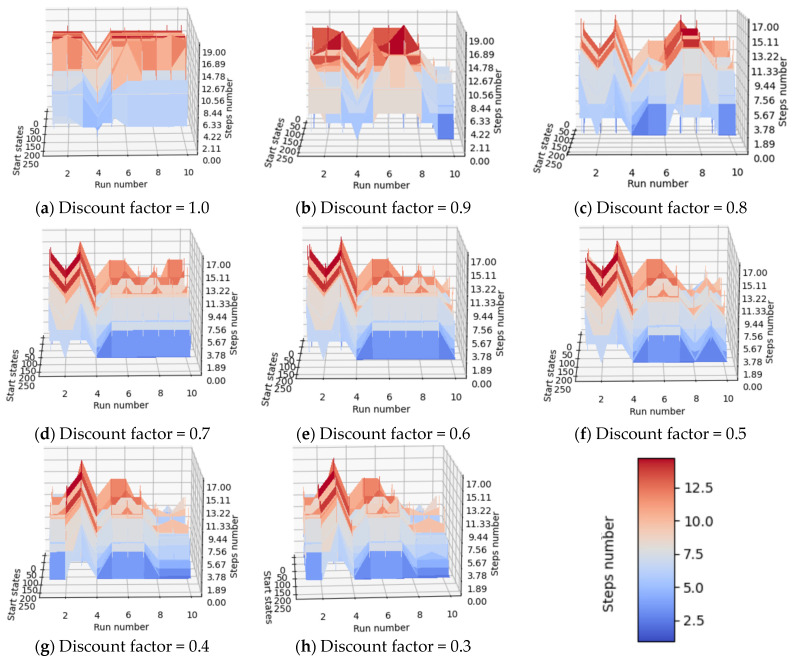
Models of trained agents vs. available initial states vs. number of steps required for agents to reach the desired final state starting from a given state.

**Figure 11 micromachines-13-01887-f011:**
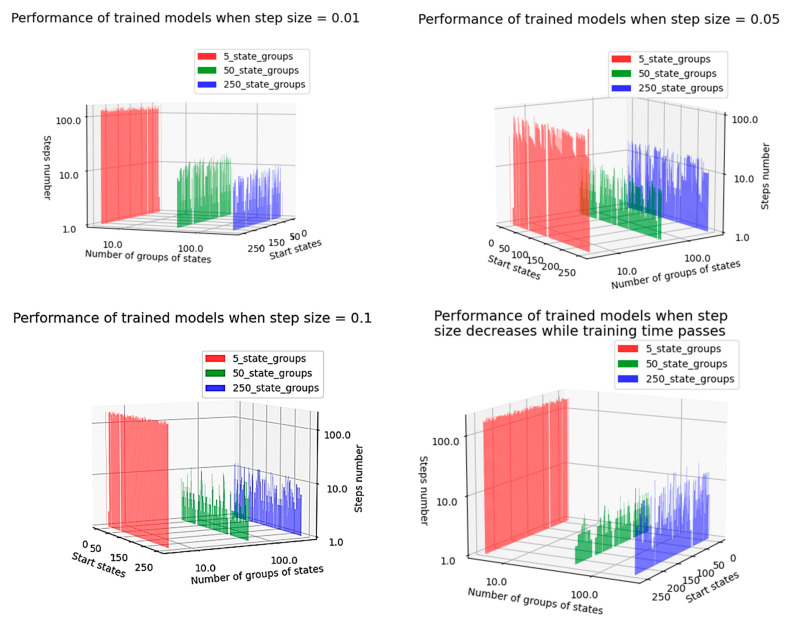
Influence of state grouping on agents’ training efficiency for several step-size settings.

**Figure 12 micromachines-13-01887-f012:**
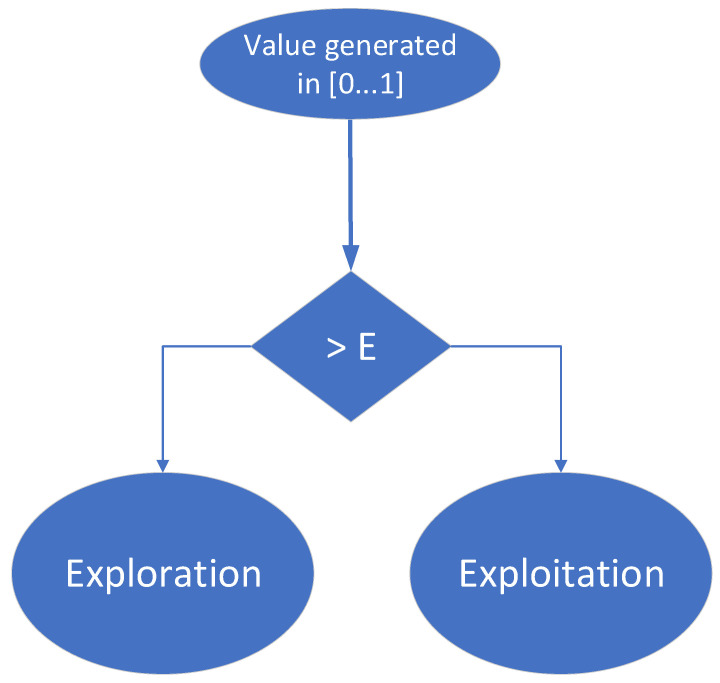
Decision made by agents during each training stage while using ϵ-greedy policy.

**Figure 13 micromachines-13-01887-f013:**
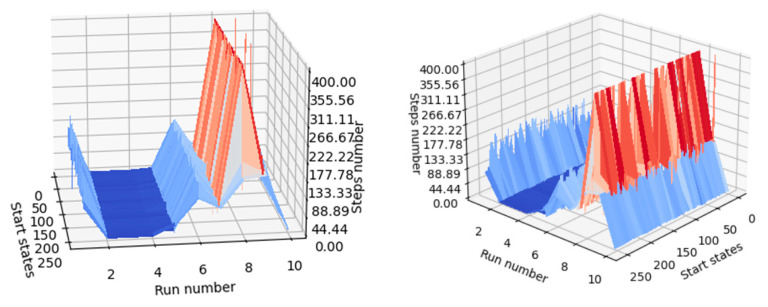
Visual indicator of agent performance (same graph observed from two positions).

**Figure 14 micromachines-13-01887-f014:**
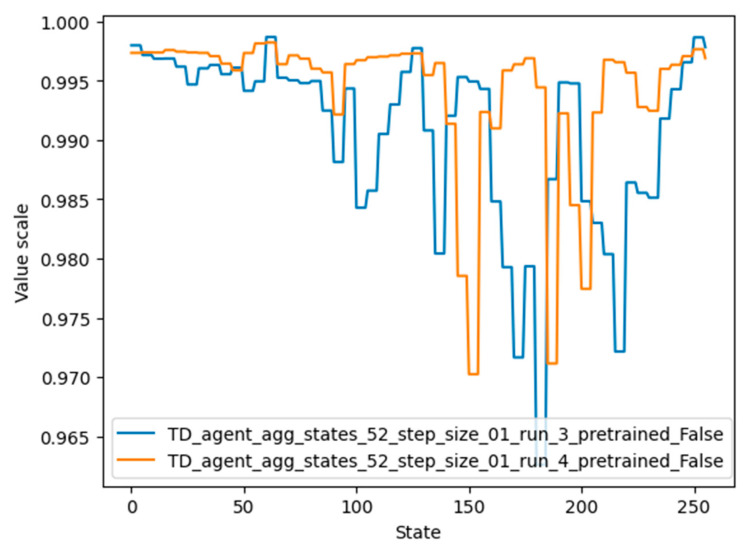
Values for states of the environment, learned by agent numbers 3 and 4 (the best-performing agent).

**Figure 15 micromachines-13-01887-f015:**
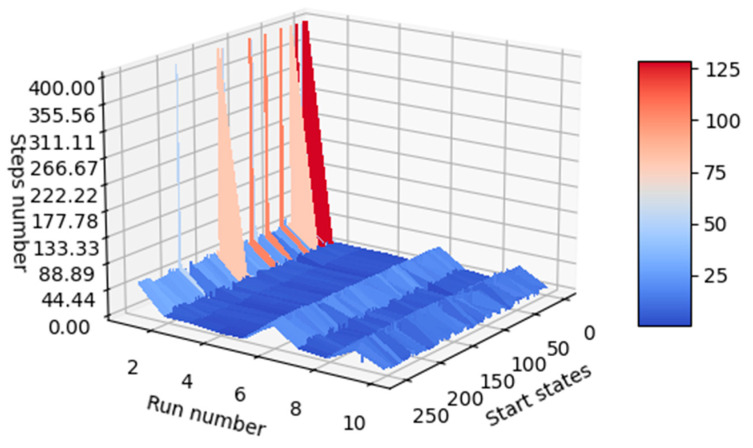
Comparison of the performance of newly trained agents.

**Figure 16 micromachines-13-01887-f016:**
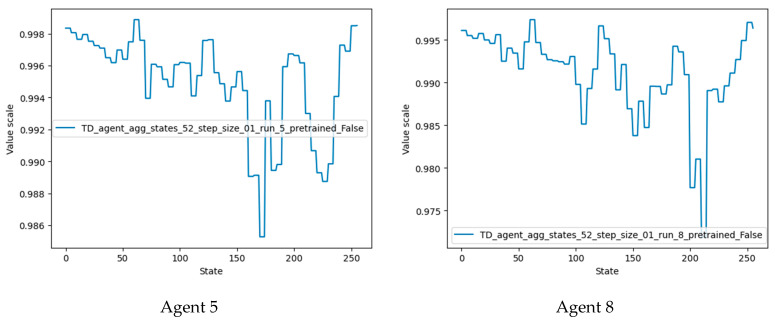
State-value plots for the best-performing agents.

**Table 1 micromachines-13-01887-t001:** Differences between related works.

Paper	Aims	Main Advantages	Drawbacks	Algorithm Used
[12]	Functional coverage fulfillment; near-miss tracking; discovering errors by focusing to reach certain functional states of the DUT	Focusing on several aspects of FV	Few details on RL-based algorithms’ configuration	Q-learning
[17]	State transition coverage fulfillment	Use of ML algorithms to increase the performance of an RL approach	Only a type of coverage in focus (state transition coverage) and only one type of device in focus (memories)	K-NN, Bayesian optimization
[18]	Functional coverage	The library created allows the use of many RL algorithms without the need to know many of their implementation details	Need to use cocotb testbenches instead of widely used ones (e.g., RTL-based testbenches)	Soft actor critic
[21]	Reaching a target state of a DUT	The degree of automation is increased by analyzing the results provided by RL algorithms using deep neural networks	The proposed approach requires many computational resources	Q-learning
[22]	Functional coverage fulfillment	Using recurrent neural networks for learning DUT behavior	The algorithms used are very complex, and the time required for verification engineers to be comfortable with them could be significantly large	Rainbow agent, based on Q-learning
Current work	Reaching a target state of a DUT	Detailed presentation of configuration possibilities for RL-based systems; development of efficient inference mechanisms	Not using ML algorithms; not working with Q-learning, which is the widely used option due to its proven performance	Semigradient temporal-difference (SGTD)

**Table 2 micromachines-13-01887-t002:** External module signals and their description.

Signal Direction	Signal Name	Signal Description
Inputs	clock	system clock
	reset_n	reset active in “0”
	operand A	first operand of chosen operation
	operand B	second operand of chosen operation
	operation code	00-division, 01-addition, 10-subtraction, 11-multiplication
	trigger	trigger that starts chosen operation
Output	result	result of performed operation over operands

**Table 3 micromachines-13-01887-t003:** Performance comparison between trained agents with a similar configuration and the untrained agent (visible in the random generation column).

	The Number of Steps Required to Reach the Target for Agents Trained			
Start State	Agent 1	Agent 2	Agent 3	Random Generation
248	2	2	5	237
34	4	4	4	41
144	11	10	12	160
50	9	10	12	51
0	3	3	3	58
90	9	8	10	126
45	10	9	11	254
255	2	2	2	5
15	3	3	6	27
69	14	15	17	34
199	7	7	10	212
Steps Average:	6.72	6.63	8.36	109.55

**Table 4 micromachines-13-01887-t004:** Comparison between agents trained using different reward schemes.

Start State	Number of Steps Required to Reach the Target for Agents Trained Using:				
	First Reward Scheme with MVS	Second Reward Scheme with MVS	Third Reward Scheme with MVS	Random Operation at Each Step	Only Addition or Subtraction
248	5	55	5	141	120
34	4	21	4	303	15
144	12	28	3	92	26
50	12	19	9	114	37
0	3	13	18	136	20
90	10	>256	10	102	13
45	11	19	11	35	78
255	2	>256	2	218	3
15	6	25	6	56	66
69	17	22	7	195	21
199	10	29	6	105	37
Steps Average:	8.36	Large	7.36	136.09	39.64

**Table 5 micromachines-13-01887-t005:** Values of training parameters for the results in Table 4.

Parameter	Value
Desired terminal states	61, 62, 63
Number of training episodes	3000
Number of groups	50
Width of operand A	8 bits
Discount factor	1
Step size	0.01
Number of reachable states	250
Operand B value	2

**Table 6 micromachines-13-01887-t006:** Configuration parameters are held constant during the evaluation of the effect of the discount factor value on the agent training quality.

Parameter	Value
Desired terminal states	61, 62, 63
Dangerous terminal states	n/a
Number of training episodes	3000
Number of groups	52
Width of operand A	8 bits
Step size	0.01
Number of reachable states	256

**Table 7 micromachines-13-01887-t007:** Influence of the discount factor on the quality of agent training.

		Steps Number ^1^	Label
**Discount** **Factor** **(γ)**	0.3	4.6	run_10
	0.4	4.2	run_8
	0.5	5.5	run_10
	0.6	4.6	run_4
	0.7	4.6	run_4
	0.8	4.7	run_4
	0.9	6.5	run_9
	1.0	8.4	run_4

^1^ Average number of steps required for each best-performing agent (evaluation was performed separately for each discount factor) trained 3000 episodes to reach the desired end state from each available state.

**Table 8 micromachines-13-01887-t008:** Configuration parameters held constant throughout the training process in the Section 4.5.

Parameter	Value
Desired terminal states	61, 62, 63
Dangerous terminal states	1, 2, 3
Operand B possible values	2, 3, 4
Width of operand A	8 bits
Discount factor	1
Number of training episodes	2000
Number of reachable states	256

**Table 9 micromachines-13-01887-t009:** Effect of state grouping on training performance when inference was performed using the MVS approach.

		Number of Steps Required to Reach the Target for Agents Trained in Following Configuration:					
Number of Groups		5		50		250	
Step-size value		$\alpha_{n}\left(a\right)=0.1$	$\alpha_{n}\left(a\right)=\frac{1}{n}$	$\alpha_{n}\left(a\right)=0.1$	$\alpha_{n}\left(a\right)=\frac{1}{n}$	$\alpha_{n}\left(a\right)=0.1$	$\alpha_{n}\left(a\right)=\frac{1}{n}$
Start State	248	159	48	12	3	5	84
	34	171	41	3	3	6	5
	144	160	43	2	2	4	32
	50	162	52	4	4	5	6
	0	169	47	7	4	12	5
	90	159	40	3	8	4	5
	45	159	41	3	8	4	5
	255	178	47	7	1	12	1
	15	174	48	2	7	2	5
	69	175	50	3	3	3	2
	199	165	49	3	3	6	2
Steps Average		166.45	46.00	4.45	4.18	5.73	13.82

**Table 10 micromachines-13-01887-t010:** Parameters used to configure the training processes for the 10 agents in focus in this section.

Parameter	Value
Desired terminal states	61, 62, 63
Dangerous terminal states	n/a
Operand B possible values	2, 3, 4
Width of operand A	8 bits
Discount factor	1.0
Number of training episodes	3000
Number of reachable states	256
Step size	0.1
ϵ initial value	1.0
ϵ decrement	0.00001
ϵ minimum value	0.01
Number of states/group	5
ϵ decrementing	On exploration branch

**Table 11 micromachines-13-01887-t011:** Details of the training process of the 10 analyzed agents.

Agent/Run Number	Training Time (m/s)	Epsilon Value at the End of the Training Process	Average Steps Number/Episode Computed in 20 Inference Episodes
**1**	00:52	0.0114	128.55
**2**	00:05	0.505	4.1
**3**	00:05	0.508	4.55
**4**	00:06	0.463	3.7
**5**	00:05	0.477	7.65
**6**	00:14	0.232	129.1
**7**	00:16	0.172	56.9
**8**	00:28	0.081	336.55
**9**	00:09	0.353	156.55
**10**	00:05	0.522	4.15

**Table 12 micromachines-13-01887-t012:** Details of the training approach when the ϵ decrease was equal to 0.000005.

Agent Number	Training Time (m/s)	Epsilon Value at the End of the Training Process	Average Steps Number Computed in 20 Inference Processes
**1**	00:08	0.573	76.55
**2**	00:07	0.606	7.8
**3**	00:05	0.658	3.5
**4**	00:05	0.677	3.25
**5**	00:06	0.673	3.05
**6**	00:05	0.668	27.95
**7**	00:07	0.615	7.2
**8**	00:05	0.648	3.05
**9**	00:06	0.676	19.6
**10**	00:06	0.647	5.4

**Table 13 micromachines-13-01887-t013:** Different training configurations and their best result after training 10 agents using each configuration.

Parameter	Configuration a	Configuration b	Configuration c
Desired terminal states	61, 62, 63	61, 62, 63	61, 62, 63
Dangerous terminal states	n/a	n/a	n/a
Operand B possible values	2, 3, 4	2, 3, 4	2, 3, 4
Width of operand A	8 bits	8 bits	8 bits
Discount factor	1.0	1.0	1.0
Number of training episodes	3000	3000	3000
Number of reachable states	256	256	256
Step size	0.1	0.1	0.1
ϵ initial value	1.0	1.0	1.0
ϵ decrement	0.000001	0.0000005	0.0
ϵ minimum value	0.01	0.01	0.01
Number of states/group	5	5	5
ϵ decrementing	On exploration branch	On exploration branch	On exploration branch
Minimum average of steps number	2.95	2.6	2.7

## Data Availability

Not applicable.

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
