# Peer review of "Reinforcement Learning Made Affordable for Hardware Verification Engineers"

_micromachines, 2022, doi:10.3390/mi13111887_

Round 1

Reviewer 1 Report

This article introduces a good and sound application of reinforcement learning. The details sound very useful to fresh RL practitioners with the useful details.  However, the article needs to be revised in order to make it neath and ready for publication, where I suggest to pay attention to the following points:

1. Abstract does not include a good justification of the study, it should be revised to include a sound and justifying sentence in which the novelty of using RL for the problem in hand is made apparent.

2. There is a commonly missed out reference across the text, please make sure "Error! Reference source not found" are resolved with providing the relevant references. 

3. The approach is time to time referring to the code used to delivery functionalities, which I don't think is appropriate . Please make use of Python hidden to the readers since this is nothing to do with Python language, but, it is a RL application. 

4. Conclusions can be made more concise and articulate. A reference is provided in Conclusions session, which I think is not appropriate, please just remove it.  

5. I can see the authors refer to equations as Formula, please replace that with Equation to make it more technically sound. 

Reviewer 2 Report

I read with pleasure your paper on RL for hardware verification engineers. It is generally well written in a good English. 

I am not extremely expert in hardware design but I have some suggestions to make the paper achieve the desired goals better, at least in my opinion.

The main goal of the paper is to make adaptation of such methods between engineers (that may have not know RL) easier. In this regard, although the explanations are very detailed, it fails a bit. Without any knowledge of RL it is impossible to understand and follow the discussion especially at the beginning. No clear explanation are given about which RL algorithm is really used, how RL works (at least in its most basic form) to make the idea clear for engineers.

For example Section 2.2 starts and uses concepts lie the "state-value function" without explaining its meaning. Someone without RL background will have no chance of understanding. 

I suggest to the authors to explain RL ideas and concepts more in detail, or at least point to material that the reader _must_ read in advance to be able to follow the many pages of discussions.

I also suggest to the authors to clarify more what kind of problem they are trying to solve with RL. For example they talk about states and values (62,63,etc.) but is not clear what they mean. What problem are the authors describing? What test are they describing? Does the described method apply to all IC?

I think those modifications are necessary, otherwise the paper remain a bit obscure and will not reach its goal of making the adoption of such methods easier.

Much more clarity is needed, in both explaining RL to engineers in the initial sections (see for example 2.2), and to explain exactly what the environment looks like (in RL terminology), what the rules are, which algorithms are trained and so on.

Additionally the authors should check the references to figures, as they are missing (an error is printed in their place).

Reviewer 3 Report

1. The contribution of the paper looks minimal in the Introduction. Clearly emphasize what is the contribution of the paper. 2. The related work section is not there in the paper. Currently, only the theoretical background section is provided with limited information. Create a section on related work and compare the work. A table will be beneficial to understand the advantages and disadvantages of the previous work and your work. It is important to emphasize what is the need for your method. 3. Reinforcement learning is well-studied for similar kinds of problems. What difference is brought by your work? 4. Discuss the novelty of the work. 5. Simulation platform is not discussed well. 6. The result section is not written well. First, it is too long (17 pages long). Second, the information presented is not clear.

Round 2

Reviewer 2 Report

Thank you for expanding the paper. I think now it contains enough information for publication. As a minor note please check the parenthesis in Formula 1, as they looks strange. But probably the editing team will take care of it.

Reviewer 3 Report

Most of my comments are reasonably addressed. I would propose to accept the paper.